# Pharmacological Support for the Treatment of Obesity—Present and Future

**DOI:** 10.3390/healthcare11030433

**Published:** 2023-02-02

**Authors:** Marcin Kosmalski, Kacper Deska, Bartłomiej Bąk, Monika Różycka-Kosmalska, Tadeusz Pietras

**Affiliations:** 1Department of Clinical Pharmacology, Medical University of Lodz, 90-153 Łódź, Poland; 2Students’ Scientific Association Clinical Pharmacology, Medical University of Lodz, 90-153 Łódź, Poland; 32nd Department of Psychiatry, Institute of Psychiatry and Neurology in Warsaw, 02-957 Warszawa, Poland; 4Department of Electrocardiology, Medical University of Lodz, 92-213 Łódź, Poland

**Keywords:** obesity, obesity treatment, metabolic syndrome, present and future of obesity treatment

## Abstract

Obesity is a growing civilization problem, associated with a number of negative health consequences affecting almost all tissues and organs. Currently, obesity treatment includes lifestyle modifications (including diet and exercise), pharmacologic therapies, and in some clinical situations, bariatric surgery. These treatments seem to be the most effective method supporting the treatment of obesity. However, they are many limitations to the options, both for the practitioners and patients. Often the comorbidities, cost, age of the patient, and even geographic locations may influence the choices. The pharmacotherapy of obesity is a fast-growing market. Currently, we have at our disposal drugs with various mechanisms of action (directly reducing the absorption of calories—orlistat, acting centrally—bupropion with naltrexone, phentermine with topiramate, or multidirectional—liraglutide, dulaglutide, semaglutide). The drugs whose weight-reducing effect is used in the course of the pharmacotherapy of other diseases (e.g., glucose-sodium cotransporter inhibitors, exenatide) are also worth mentioning. The obesity pharmacotherapy is focusing on novel therapeutic agents with improved safety and efficacy profiles. These trends also include an assessment of the usefulness of the weight-reducing properties of the drugs previously used for other diseases. The presented paper is an overview of the studies related to both drugs currently used in the pharmacotherapy of obesity and those undergoing clinical trials, taking into account the individual approach to the patient.

## 1. Introduction

Obesity is a growing civilization problem of the developed and developing countries of the twenty-first century. The World Health Organization estimates that, globally, 605 million people are obese and more than 1.9 billion adults are overweight [1,2]. The available data indicates that the number of obese patients has doubled since 1980. The obesity rate has dramatically increased in both males and females, and across all ages, with proportionally higher prevalence in elderly persons and women. While this trend is present globally, the absolute prevalence rates vary across regions, countries, and ethnicities. The incidence of obesity also varies with socioeconomic status, with slower rates of Body Mass Index (BMI) increase in high-income and some middle-income countries [3].

Central obesity is one of the most important components of metabolic syndrome (MS) and is associated with a number of life-threatening complications, deterioration of its quality, disability, and the risk of developing metabolic disorders (including insulin resistance–IR, type 2 diabetes–T2DM, non-alcoholic fatty liver disease–NAFLD, cardiovascular diseases (including arterial hypertension-AH, myocardial infarction, stroke). However, many other diseases, for example musculoskeletal disease (osteoarthritis), Alzheimer disease, depression, obstructive sleep apnea (OSA), and some types of cancer (for example, breast, ovarian, prostate, liver, kidney, and colon) increase with the presence of excess body weight. In addition, obesity might lead to a reduced quality of life, unemployment, lower productivity, and social disadvantages [4,5,6,7,8,9,10]. 

In view of the above clinical picture, obesity and its treatment pose a great challenge to modern medicine. Taking into account the recommendations of the international societies dealing with the combat against obesity, it is necessary to first take into account all the individual approaches to the patients, their age, gender, comorbidities, and organ capacity, as well as socioeconomic status [11,12,13]. The appropriate behavioral methods, based on changing the diet and increasing physical activity, constitute the basis of the therapeutic possibilities in obesity. Studies evaluating the effectiveness of various types of diets, including vegan, very low-calorie ketogenic, and Mediterranean diets are available. The effectiveness of the use of a low-calorie diet with balanced proportions of the main nutrients, such as carbohydrates, proteins, and fats, in order to achieve a negative energy balance in the dietary treatment of obesity has been demonstrated [14]. Introducing these changes is a very difficult and long process, as it often requires a major commitment of the patients and a change in their current habits [15]. However, even after achieving the desired body weight, it is a significant problem to maintain it at a reduced level [16].

With regards to bariatric surgery, it is considered the most effective treatment to date [17]. However, despite its great efficacy, it is associated with a number of negative outcomes (perioperative chyloperitoneum and chylothorax) and postoperative complications (anemia, depression, fractures, malabsorption, etc.) [18,19,20,21]. Therefore, due to the lack of desirable or definite weight loss, it raises a lot of controversy and concern among patients with obesity [20,22]. Not without significance is also the issue of financing this type of treatment [18,22]. The limitations with bariatric surgeries, along with the lifestyle modifications difficulties, has resulted in growing interest in the development of obesity drugs not only among the pharmaceutical industry but also with patients and the practitioners. Currently, we have a growing selection of slimming drugs, some of which are still in clinical trials. They differ both in the mechanisms of action, effectiveness, safety, and the possibility of application in a particular patient. Therefore, the purpose of the following review is to summarize the latest evidence about weight-reducing drugs, taking into account their effectiveness and the individualization of use. 

## 2. Drugs Registered for Obesity Treatment

### 2.1. Orlistat

Orlistat (tetrahydroxylipostatine) is one of the few drugs registered for the treatment of excess body weight in patients with BMI ≥ 30 kg/m^2^ or ≥28 kg/m^2^, who are at risk of illness because of their weight [23]. It is a potent, specific, irreversible inhibitor of food lipases (gastric and pancreatic), which, by reducing the absorption of fats from the gastrointestinal tract, limits the calorie content of the consumed meals. It exerts its pharmacological effect by forming a covalent bond with the active serine site of gastric and pancreatic lipases in the lumen of the gastrointestinal tract. This action prevents these enzymes from hydrolyzing dietary fat (in the form of triglycerides–TG) into absorbable free fatty acids and monoglycerols. The agent does not seem to inhibit the activity of other pancreatic enzymes, and its absorption into the systemic circulation is insignificant. However, its minimal effect in a situation of low fat intake should be emphasized (<45 g or <20% energy from fat daily), but otherwise, the percentage of the dietary fat excreted with the orlistat use increases in a dose-dependent manner, plateauing at around 30–35% for dosages of 180–360 mg per day, compared with a normal fecal fat excretion of 5%. When taken orally, orlistat is almost entirely excreted via the feces within 3–5 days. There is minimal systemic absorption and no accumulation during at least 2 years of the treatment. Orlistat is indicated for the treatment of obesity or those who are overweight in the presence of associated risk factors, in conjunction with a hypocaloric diet. The dosage of orlistat is 180–360 mg per day, in three divided doses within, or up to, 1 h after each main meal. Patients should follow a nutritionally balanced hypocaloric diet containing no more than 30% calories from fat (<67 g fat per day for a 2000 kcal diet), and the fat intake should be distributed evenly across each meal to minimize the gastrointestinal adverse effects. If a meal does not contain fat, the dose of orlistat can be omitted. It is also advised that patients take a multivitamin supplement containing fat-soluble vitamins, administered at least 2 h before or after the administration of orlistat. Orlistat is contraindicated in pregnancy and lactation, and in people with chronic malabsorption syndromes or cholestasis [24,25].

The efficacy of orlistat in weight reduction in obese patients was demonstrated many years ago in two large, randomized, multicenter clinical studies. The first study conducted by Sjostrom, with the use of this drug at a standard dose of 120 mg three times a day together with a slightly low-calorie diet (600 kcal/day deficiency) over about a year resulting in approximately 10% reduction in the body weight (10.3 kg). It is noteworthy that, over the next year, the patients who continued to take orlistat regained, on average, half their body weight as compared with the patients who switched to the placebo [26]. The study conducted by Davidson on the use of the same dose of the drug for a year allowed for slightly smaller but also statistically significant weight reductions (8.76 ± 0.37 kg), with patients using 360 mg of orlistat per day regaining less body weight (approximately 35%) than those using a dose of this drug reduced by half (approximately 51%) or the placebo (approximately 63%) [27].

In a study comparing the effectiveness of orlistat at a daily dose of 360 mg with liraglutide up to 3 mg/day for 7 months in a group of obese and overweight patients, not only was a significant reduction in body weight found when using both drugs, but also in fasting plasma glucose (FPG), systolic blood pressure (SBP), low-density-lipoprotein cholesterol (LDL-C), and alanine transaminase (ALT). The reduction in body weight with liraglutide (7.7 kg) was significantly greater than that observed with orlistat (3.3 kg), and more individuals lost at least 5% of their baseline weight with liraglutide (64.7%) than with orlistat (27.4%). The rates of prediabetes significantly decreased with liraglutide in comparison to orlistat [28].

When considering therapy, the choice of treatment in a special population should be kept in mind. In addition to significant weight reduction, this drug also has additional properties and may be preferred in the treatment of obesity in patients with NAFLD and polycystic ovary syndrome (PCOS). The use of this drug at 120 mg per day for 12 weeks has been shown to significantly reduce the triglyceride-glucose index and free fatty acid (FFA) content in patients with NAFLD [29]. In addition, it is suggested that it may not only reduce the IR and fat content of the liver, but also prevent its fibrosis in this group of patients [30]. The usefulness of orlistat was also confirmed in overweight and obese patients using ethinyl-estradiol/drospirenone due to PCOS, in whom the addition of orlistat to that therapy was associated with a reduction in lipoprotein(a), an independent risk factor predicting an increased risk of CVDs [31]. In turn, Zhao et al. proved the greater effectiveness of orlistat in the improvement of polycystic ovarian morphology and the similarity to metformin in terms of weight loss and improvement of IR, lipid metabolism, and menstrual cycle in overweight/obese PCOS patients [32]. The addition of this medicine to drospirenone/ethinyl estradiol therapy in overweight or obese patients has also been demonstrated to significantly affect the reduction in visceral fat area, body weight, hip circumference, BMI, level of free testosterone, LDL-C, FPG and blood pressure (BP) [33]. The meta-analysis conducted by Chen et al. also showed that this drug, when used by obese or overweight patients with PCOS taking oral contraceptives, has a positive effect on not only the anthropometric parameters, but also on lipid metabolism (TC, LDL-C, and TG levels were reduced and the high-density lipoprotein cholesterol–HDL-C level was increased), carbohydrate (fasting serum insulin–FINS and Homeostatic Model Assessment–Insulin Resistance-HOMA-IR levels were reduced), and hormone levels (testosterone, sex hormone-binding globulin, free androgen ratio, lutropin, dehydroepiandrosterone, follicletropin, and estradiol). In addition, a significantly higher rate of ovulation and pregnancy was observed in the group of patients using orlistat [34]. It is also suggested that this drug may reduce inflammation (reducing the concentration of such cytokines as tumor necrosis factor delta and interleukin-6) [35,36,37] and the incidence of T2DM [38,39], and also may be used as an adjunctive therapy for lipoprotein lipase deficiency in adults, given its availability and favorable safety profile [40]. Available research suggests its usefulness in oncological therapy, through mediated FASN inhibition it could overcome sorafenib resistance and enhance the cell killing in hepatocellular carcinoma (HCC) by changing the cell metabolism [41,42]. A recently published propensity-score matched cohort study found that the use of orlistat for about 6 years is associated with a reduction in mortality, myocardial infarction incidents, ischemic stroke, new-onset heart failure, and chronic kidney disease stage III [43].

### 2.2. Bupropion with Naltrexone

The combination of bupropion with naltrexone is another therapeutic option for patients with a BMI ≥ 30 kg/m^2^ or ≥27 kg/m^2^ and at least one comorbidity, such as AH, T2DM, or dyslipidemia [44]. It is also registered for the treatment of depressive disorders and nicotine addiction [45]. Bupropion is a selective inhibitor of catecholamines (norepinephrine and dopamine) reuptake, affecting the turnover of norepinephrine throughout the body. It stimulates the hypothalamic proopiomelanocortin (POMC) neurons releasing the alpha-melanocyte stimulating hormone (alpha-MSH), which binds to the melanocortin 4 receptor. The stimulation of this receptor initiates a cascade of actions that results in a decrease in the energy intake and an increase in its expenditure [46]. In addition to alpha-MSH, the POMC neurons secrete beta-endorphins, which are endogenous agonists of the opioid receptor. The beta-endorphins are involved in negative feedback on the POMC neurons, inhibiting the excretion of α-MSH. The opioid receptor’s blockade by naltrexone inhibits the pathway of the negative feedback, allowing for stronger and more permanent activation of POMC neurons, and consequently increasing the anorexic effect of bupropion, thus giving a hyperadditive effect [47]. Naltrexone, on the other hand, is a long-acting, specific antagonist of opioid receptors with an unknown agonistic component. It binds competitively to the receptors located in the central and peripheral nervous system, blocking the access of exogenous opioids [48]. Taking advantage of the above mechanism, it is used in the treatment of alcohol and opioid dependence, as well as in severe opioid poisoning [49,50]. In the research of Burns et al. on obese animal models, the combination of bupropion with naltrexone was demonstrated to reduce food intake and body weight, increase dopaminergic receptor D2 (Drd2) expression in rats on a high-fat diet, and increase tyrosine hydroxylase levels in rats on both a high-fat and standard diet. The level of dopaminergic receptors D1a remained unaffected. This may indicate that weight loss is associated with reduced food intake in response to an increased expression of Drd2 in the mesolimbic areas of rats that received a high-fat diet [51].

The effectiveness of naltrexone and bupropion combination therapy has been proven both in the group of healthy patients and in those with comorbidities such as AH, T2DM, and dyslipidemia (Table 1).

The analysis results of the Contrave Obesity Research (COR) program, except for the COR-DM clinical trial, showed that people who reached the 16-week threshold of about 5% weight loss during the treatment are likely able to maintain clinically significant weight loss after the first year of the therapy [56]. Additionally, this program demonstrated, this time without excluding patients with COR-DM, that weight loss, independent of treatment, was associated with improved ALT and Fibrosis-4 (FIB-4) index. However, a significant independent effect of naltrexone/bupropion on the change from baseline was observed for FIB-4, but not for ALT. The categorical ALT response (from above to within the normal ranges: 10–40 IU/L for men; 7–35 IU/L for women) and the achievement of 25% and 50% reduction in ALT were greater in the case of the naltrexone/bupropion treatment, and independently affected by weight loss, but not treatment. It should be emphasized that these conclusions are not applicable to patients with ALT and an aspartate aminotransferase (AST) activity greater than 2.5 times the upper limit of normal, who were not assessed in that study [57].

Hypothalamic structures are associated with the pathogenesis of binge eating disorder (BED). Consuming high-calorie meals is becoming an increasingly recognized problem in mental disorders, and obesity may be the result. A study by Carbone et al. proved that the use of a prolonged action preparation combining naltrexone with bupropion in obese patients with BED not only significantly reduces the body weight (85% of patients lost at least 5% of their initial weight), but also improves pathological eating behavior. However, it should be noted that this study included a small sample size, so these conclusions should be interpreted with extreme caution [58].

Considering the institution of this drug combination in a particular patient, it should be remembered that it should not be used in patients with seizure history, eating disorders, or those undergoing ethanol. In addition, they must not be combined with other CNS depressants, in particular opioids. The drug is also contraindicated in people with a history of hypersensitive reactions to naltrexone or any other component of the preparation [59,60].

### 2.3. Phentermine with Topiramate

Phentermine is a central sympathomimetic leading to increased secretion of serotonin, norepinephrine, and dopamine, whereas topiramate is a gamma-aminobutyric acid (GABA) agonist, glutamate antagonist, and carbonic anhydrase inhibitor. Although the exact mechanism of this combination is still unknown after many years of research, it is thought that it may reduce food intake. While this combination was approved by the US Food and Drug Administration for patients with an initial BMI of ≥30 kg/m^2^ or ≥27 kg/m^2^ in the presence of at least one weight-related comorbidity, such as AH, T2DM, or dyslipidemia, it has not been approved by the European Medicines Agency (EMEA) due to ongoing safety and efficacy concerns [61].

The effectiveness of this drug combination has been proven in four randomized clinical studies involving overweight and obese patients (Table 2).

Data from available studies indicate significant benefit in weight reduction from this combination in adults with compulsive overeating syndrome or bulimia nervosa. Phntermine/topiramate combination significantly reduced the number of objective binge eating days and period in 28 days surprisingly without a significant effect on the vital signs. The responses were not significantly different for binge-eating disorders versus bulimia nervosa. It should be emphasized, however, that these results concerned a study conducted on 22 patients. It should be noted that the results of this study may be limited due to the small sample size and loss of large number of participants due to safety concerns. The largest number of exclusions were for a BMI of <21 kg/m^2^, or for a prior history of anorexia nervosa [65]. In addition, there are isolated reports of benefits of phentermine with topiramate in the treatment of obesity secondary to night snacking syndrome in terms of both the weight reduction and resolution of sleep-related eating disorder behaviors. It should be emphasized that the discontinuation of the drug may result in the return of an eating disorder [66]. Due to its mechanism of action, the combination of phentermine with topiramate also seems to be effective in obese patients who show a variety of sleep problems in the form of insomnia, or sleep maintenance problems [67].

When discussing the effectiveness of drugs used in obesity, the results of a meta-analysis and review by Salari et al., in which the use of phentermine with topiramate at a 7.5 mg + 46 mg dose has been shown to be associated with the highest effectiveness in weight reduction, compared to drugs such as pramlintide, the combination of naltrexone with bupropion mentioned above, liraglutide, lorcaserin, and orlistat. It should be emphasized that pramlintide is highly effective, which has not yet been registered for weight loss support treatment [68]. Pramlintide is an analogue of human amylin, FDA approved, and used along with insulin in patients with T2DM and T1DM. In addition to regulating glucose, Pramlintide increases satiety and thus reduces calorie intake through the central mechanism. It also facilitates moderate weight loss in obese or overweight patients with and without DM [69]. Certainly, it should be remembered that this drug should not be used in patients with glaucoma, a history of hyperthyroidism, and in those who have recently used a monoamine oxidase inhibitor, as it can lead to a hypertensive crisis [70].

### 2.4. Lorcaserin

Lorcaserin is a selective agonist of the 5-HT 2c serotonin receptor. It has been approved by the FDA for long-term use, combined with lifestyle modification, to promote weight reduction in obese people with or without T2DM. It should be noted that, as in the case of the combination of phentermin/topiramate, the EMEA did not approve this drug due to safety concerns [61]. Lorcaserin reduces body weight by promoting the feeling of satiety mediated by the 5-HT 2c receptor on the POMC/CART neurons in the arcuate nucleus of the hypothalamus [71]. Additionally, lorcaserin has been suggested to act via the same receptor in the stimulation of the reward system [72]. Four RCTs in which a significant effect of lorcaserin on weight reduction was observed in adults have been conducted [73,74,75]. The results of these studies are presented in Table 3. Lorcaserine has proven to reduce the incidence of DM by 19% in pre-diabetics vs. 23% in patient without DM. Additional benefits were seen in normoglycemia (non-significant) in pre-diabetics; diabetic microvascular complications (a composite of incident persistent microalbuminuria, diabetic retinopathy, or diabetic neuropathy) were reduced by 21% in diabetic patients and improved the quality of life regardless of the dose. Furthermore, [73,74,75,76] emphasized that the considerable efficacy of lorcaserine was seen not only during the first but also the second year of treatment [76]. A meta-analysis report by Sigh and Singht, revealed a fairly low therapeutic efficacy and lack of response in a large percentage of the population treated with lorcaserine [77]. It should be emphasized that the FDA recommends discontinuing lorcaserin if at least 5% of weight loss has not been achieved within the initial 12 weeks [78]. It should be remembered to exercise extreme caution among patients using drugs that affect the system of serotonergic neurotransmitters, diabetics (risk of hypoglycemia and risk factors for damage to the heart valves) [79,80].

The substances contained in food, such as carbohydrates, amino acids, and fatty acids, interact with the specific L and K cells of the ileum and colon epithelium. In response to this interaction, the L cells of the epithelium secrete glucagon-like peptide 1 (GLP-1) and the K cells secrete glucose-dependent insulinotropic peptide (GIP). In addition, GLP-1 has been demonstrated to be a stronger stimulator of the beta cells than GIP, inhibiting glucagon secretion at the same time. The GLP-1 receptor (GLP-1R) is a member of the B family of G protein-coupled receptors. The interaction between GLP-1 and GLP-1R activates the adenyl cyclase-dependent pathway and through this it is involved in controlling the insulin secretion from pancreatic B cells. Therefore, the analogues of this receptor were originally registered for the treatment of hyperglycemia in patients with T2DM. Research studies have demonstrated their additional benefits, including in patients with cardiovascular disorders, Alzheimer’s disease, Parkinson’s disease, PCOS, NAFLD, non-alcoholic steatohepatitis—NASH, tumor disease, and, first of all, obesity [81]. The widespread utility of these drug is primarily due to numerous mechanisms of action. In addition to the pancreas, GLP-1R is present in various tissues including the lungs, kidneys, nervous, cardiovascular, and digestive systems. The weight loss associated with GLP-1R analogues is thought to be achieved through a variety of mechanisms, including delayed gastric emptying, increased satiety, and increased resting energy expenditure, as well as direct effects on the appetite center of the brain [82,83]. Different signal paths lie at the root of these effects. Among other things, GLP-1 analogues have been shown to activate the Wnt signaling pathway to promote adipocyte differentiation, and to prolong the time of gastric emptying in the gastrointestinal tract. However, it should be emphasized that the inhibition of gastric emptying plays a major role in reducing postprandial hyperglycemia and is not the main mechanism for weight reduction [84,85]. Additionally, it has been proven that GLP-1 analogues can also increase the energy expenditure. Under the control of 5′AMP activated kinase in the ventromedial hypothalamus, they can promote the conversion of visceral white adipose tissue to brown adipose tissue, increasing thermogenesis and thus energy consumption [86].

Currently, there are 5 GLP-1 analogues available, including exenatide, liraglutide, lixisenatide, semaglutide, and dulaglutide. It should be emphasized, however, that this is not a homogeneous group of drugs, because they differ not only in their chemical structure, but also in length, route of administration, and elimination, as well as the risk of development of antibodies. Additional slight differences among the class include the rate of hypoglycemic and weight-reducing action, pleiotropism, and the risk of side effects. It should be emphasized that, so far, only liraglutide has been registered in the world as a drug used to support the treatment of obesity [81]. When considering these agents, one should keep in mind the contraindication for use in a patient with a personal/family history of medullary thyroid carcinoma, and in those with multiple endocrine neoplasia syndrome type 2 [87].

### 2.5. Liraglutide

Liraglutide is a short-acting GLP-1 analogue that was primarily indicated for patients with T2DM at a target dose of 1.8 mg, administered once daily subcutaneously. Since 2014 in the USA and since 2015 in Europe, its indication has been extended by up to a target dose of 3 mg once a day for patients with a BMI of ≥30 kg/m^2^ or from 27 kg/m^2^ to <30 kg/m^2^, but with at least one complication associated with abnormal body weight, such as prediabetes or T2DM, AH, dyslipidemia, or OSAS [88]. The efficacy of liraglutid has been established in four randomized clinical studies, referred to as SCALE (Satiety and Clinical Adiposity Liraglutide Evidence), the results of which are presented in Table 4. It should be emphasized that in the SCALE Obesity and Prediabetes study, liraglutide appeared to be less effective in patients with a lower BMI. The positive effects of liraglutide on HbA1c, FPG, and glucose levels during oral glucose tolerance test were greater in patients with pre-DM than in those without the condition. The prevalence or pre-DM incidences were significantly lower in the liraglutide group than in the placebo group. T2DM developed in more patients in the placebo group than in the liraglutide group during the course of the treatment [89]. In the SCALE DM RCT, liraglutide 3.0 mg was statistically significantly better than liraglutide (1.8 mg) on weight-related measures, including mean weight loss, 5% or more and more than 10% weight loss responders, as well as WC and BMI. Liraglutide (3.0 mg) was also statistically significantly better than liraglutide (1.8 mg) on glucose-related parameters (HbA1c, FPG, fasting proinsulin, proinsulin-insulin-ratio), HOMA IR and net use of oral hypoglycemic agents [90]. In turn, the SCALE Sleep Apnea study demonstrated that liraglutide 3.0 mg per day was associated with a significantly higher apnea-hypopnea index (AHI) value compared to the placebo. It should also be emphasized that. in the above study. the treatment effect on AHI did not depend on participants’ gender, baseline BMI, or OSAS severity category [91].

The studies suggest that the benefits of liraglutide extend beyond DM and includes women with PCOS. It has been demonstrated that the use of this drug, liraglutide, at a dose of 3 mg per day for 32 weeks has resulted in not only a significant weight loss, but also free androgen index, an indicator of androgenicity, and an increase in the frequency of menses. It was further reported that menses were more regular with the mean frequency of menstrual cycles [93]. A meta-analysis by Ge et al. patients with PCOS showed similar therapeutic efficacy compared with metformin with a higher weight reduction. On the other hand, the addition of liraglutide to metformin allows for a greater reduction in body weight, WC, FPG, and FISN [94]. The available data also indicates the benefits of liraglutide, similar to phentermine-topiramate and naltrexone-bupropion, in weight reduction in patients diagnosed with Prader-Willi syndrome [95].

When considering bariatric surgery, there is a possibility of lack of weight reduction and even weight gain after the procedure [96]. In the study by Elhag and El Ansari. Liraglutide administered at 3 mg per day in adult patients with inadequate weight loss or weight regain after bariatric surgery significantly reduced weight and this was effective in both groups. Liraglutide did not prove to have a positive effect on cardiometabolic risk factors (SBP, DBP, HR, FPG, HbA1c, TCH, TG, LDL-C, HDL-C, AST, ALT) in patients after the first procedure, while in patients after the second procedure, only a significant reduction in SBP was found [97]. The appealing weight loss with liraglutide 3.0 mg applies to Roux-en-Y gastric bypass, laparoscopic gastric banding, and laparoscopic sleeve gastrectomy procedures [98]. Additionally, a possible favorable effect of this drug on metabolic disturbances and on myocardial fibrosis, as well as cardiovascular death in patients with carbohydrate metabolism disorders of various severities, has been suggested [99].

### 2.6. Semaglutide

Semaglutide is a long-acting GLP-1 analogue that has been approved by both the FDA and EMEA to support obesity treatment in patients with a BMI of ≥30 kg/m^2^ or ≥27 kg/m^2^ with at least one weight-related ailment [100]. Its effectiveness in weight reduction has been demonstrated in a series of studies referred to STEP (Semaglutide Treatment Effect in People with Obesity). The results of these studies are presented in Table 5.

STEP studies have demonstrated that semaglutide 2.5 mg over 68 weeks lead to a reduction in the body weight by >20% in as many as 13.1–36.6% of patients, and 1.0 mg dose in 4.7% of patients. Moreover, there was a positive effect on physical functioning normoglycemia in pre-diabetics with 1.7 mg [102,103,104,105,106]. To date, the results of the use of 2.4 mg per day for 104 weeks of semaglutide in obese patients, as well as the results of the STEP7 study, have not been published [ClinicalTrials.gov Identifier: NCT04251156]. The STEP 8 study compared semaglutide and liraglutide at doses of 2.4 mg per week and 3.0 mg per day, respectively, and demonstrated a significantly higher efficacy of the former drug. During 68 weeks of the therapy, an average weight reduction of 15.8% was achieved with semaglutide and 6.4% with liraglutide. The statistical differences prevailing in favor of semaglutide also concerned the beneficial effects on other CVD risk factors, such as WC, TCH, VLVL, HbA1c, FPG, and CRP. It should be emphasized that a much higher percentage of patients discontinued obesity therapy in the liraglutide group [104]. Some studies indicate a benefit of semaglutide in patients with NASH, but the results remain inconclusive [107,108]. There are few reports of the effectiveness of semaglutide in children and adolescents diagnosed with Prader-Willi syndrome. The results obtained so far suggest that this drug may be effective in inhibiting appetite and weight loss in this group of patients without significant side effects [109,110]. Isolated reports indicate the effectiveness of oral semaglutide in controlling hunger and satiety. Ginnson et al. proved in their study that reducing the feeling of hunger after semaglutide administered orally has a beneficial effect on body weight and body fat in patients with T2DM [111].

## 3. Drugs Registered in Other Disease Entities Demonstrating a Weight-Reducing Effect

### 3.1. Exenatide

Exenatide, next to liraglutide, is a short-acting analogue of GLP-1. It is currently registered for the treatment of T2DM. Shoemaker et al. reported in their studies that exenatide treatment not only reduces food intake, but also reduces the body’s total energy expenditure, disproportionate to weight loss [112]. In 2019, Jabbour et al. conducted an analysis of the effectiveness of the treatment and the effect on the body weight of exenatide, the combination of exenatide with dapagliflozin, exenatide with metformin, and dapagliflozin with metformin. The study found that the greatest degree of weight reduction was obtained with a combination of exenatide and dapagliflozin. Interestingly, the highest percentage of reduction and the highest absolute value of weight reduction were achieved by patients with a high baseline BMI [113]. Another study showed that the combination treatment with exenatide and dapagliflozin may lead to more favorable central responses to the low-calorie food stimuli, which may result in patients switching to low-calorie foods [114]. A similar study demonstrated that the combination of exenatide and metformin is more beneficial for body weight, BMI, and WC in patients with PCOS. Additionally, improvements in FPG, insulin levels, and scores in the glucose load test were observed [115]. Another study additionally showed a reduction in pre-DM incidents in PCOS patients using the combination therapy [116]. Interesting research results were obtained by Rodgers et al. studying a group of obese women. Individual variability in weight loss was observed with both exenatide and low-calorie diets and placebo. There was a trend towards a higher percentage of patients who achieved ≥ 5% weight loss with exenatide compared to diet (56% treated with exenatide, 76% treated with diet), but no significant difference in those who achieved ≥ 10% weight loss (23% of those treated with exenatide and 36% treated with diet). In both groups, greater weight loss after 3 months of treatment was a predictor of a good response status. Both groups also showed similar peak weight loss during the study period [117].

### 3.2. Dulaglutide

Dulaglutide, currently registered for T2DM, is another GLP-1 agonist. In a meta-analysis by Zhang et al., it appeared that dulaglutide alone did not significantly affect the weight of patients with T2DM, compared to placebo in 23 RCTs including 2802 patients. On the other hand, it significantly reduces the body weight as an addition to the patient’s insulin therapy, compared to the control group [118]. Retrospective studies in routine practice have reported dulaglutide in diabetic patients not meeting the criteria for inclusion in RCTs had an average weight loss of 2.9 kg (*n* = 83,116) [119]. A considerable limitation of this study was the ambiguity of the follow-up period. In addition, it was demonstrated that in patients diagnosed with BED, after 12 weeks of therapy, the number of attacks of compulsive eating decreased, and the body weight, BMI, and glycated hemoglobin levels were statistically significantly reduced during the dulaglutide therapy compared to the placebo [120]. There are currently no studies on the efficacy of dulaglutide in obese patients without T2DM.

### 3.3. Tirzepatide

Tirzepatide is a dual GLP1 and GIP receptors agonist. In 2022, tirzepatide was approved by the US FDA for the treatment of obesity [121]. Clinical studies have shown the significant effectiveness of tirzepatide in weight reduction. In a multicenter, randomized, placebo-controlled, double-blind Phase 3 study by Jastreboff et al., 2539 overweight/obese adult participants took part. Divided into four groups, they received a weekly subcutaneous injection of tirzepatide at a dose of 5 mg, 10 mg, 15 mg, or placebo. After 72 weeks, the weight reduction in each group was, respectively, 15%; 19.5%; 20.9%; and 3.1% [122]. Studies also indicate a much higher effectiveness of tirzepatide compared to semaglutide in weight reduction [123,124]. In a Phase III SURPASS-3 study of tirzepatide (5, 10, or 15 mg) in subjects with T2DM (with or without metformin and/or an SGLT-2 inhibitor), the weight reduction ranged from 9.8 to 15.2 kg [125]. In 2022, tirzepatide was registered in the US by the FDA as a drug supporting the treatment of obesity. Although the exact mechanisms of GLP-1/GIP synergism are unclear, it has been hypothesized that GIP may act directly through the CNS by inhibiting food intake, enhancing the anorexigenic effects of GLP-1, or increasing tolerance to GLP-1R agonists [126,127]. Taking into account the theoretical risks, the use of tirzepatide should be avoided in people with a personal or family history of medullary thyroid carcinoma. Patients with a history of multiple endocrine neoplasia type 2 (MEN 2) should also avoid tirzepatide. Tirzepatide is only approved for patients with T2DM and should not be used in people with T1DM or other forms of DM, such as latent autoimmune diabetes in adults. Patients who are currently using other GLP-1 analogues, such as semaglutide or liraglutide, should not be prescribed tirzepatide. Patients treated with insulin may initiate tirzepatide therapy and carefully reduce the insulin dose to minimize the risk of hypoglycemia [128,129,130]. There are also other relative contraindications, such as gallbladder disease or diabetic retinopathy [131].

### 3.4. Glucose-Sodium Cotransporter Inhibitors (SGLT-2i)

SGLT-2i inhibits glucose resorption in the renal tubules, and thus increase diuresis, glycosuria, and natriuresis. The degree of their effectiveness depends on the renal filtration rate (eGFR). Weight reduction is caused by the energy loss associated with the glucose loss and a decrease in the body fat content due to natriuresis and forced diuresis [132]. They belong to the group of drugs registered primarily for T2DM therapy, with few indicated for heart failure and kidney disease [133]. The available data on their beneficial effects on body weight are inconsistent and appear to be dependent on comorbidities and the SGLT-2i used.

A meta-analysis of studies on the effectiveness of antidiabetic drugs in weight reduction showed that subcutaneous semaglutide was most effective at reducing weight, followed by semaglutide administered orally, exenatide, liraglutide, and SGLT-2i, such as empagliflozin, canagliflozin, dapagliflozin, and ertugliflozin. Metformin had little effect on weight loss. According to the authors of the meta-analysis, SGLT2i may be a good choice of antidiabetic treatment in patients with T2DM accompanied by overweight or obesity [134]. In contrast, Janez and Fioretto reported that the benefits of hypothetical weight loss in obese patients are not reflected in clinical situations. They suggested better metabolic and weight loss effects will be attained through lifestyle modification in T2DM patients treated with SGLT2i [135].

As it follows from a meta-analysis by Zheng et al., a statistically significant reduction in absolute body weight was noted in the SGLT2i group. The authors claim that SGLT2i can be used in overweight and obese patients without diabetes, provided there is a low risk of urinary tract infection [136]. In turn, a meta-analysis of Wong et al. emphasizes that SGLT2i causes a slight decrease in body weight, without a significant reduction in WC in overweight and obese patients [137]. Therefore, it seems reasonable to use SGLT-2i, which will play an augmentative role in the treatment of obesity in patients without concomitant T2DM. SGLT2i should be considered in patients with heart failure (with reduced ejection fraction) or chronic kidney disease (with or without diagnosed cardiovascular disease), as cardiovascular data indicates that these drugs provide protection against serious cardiovascular disorders in people diagnosed with atherosclerotic cardiovascular disease, reduces the risk of hospital admission for HF, and reduces mortality due to cardiovascular as well as other reasons [138]. Particular caution should be exercised in patients with risk factors for bone fractures and limb amputations [139].

One of the meta-analyses on the effects of SGLT-2i on body weight in obese patients without concomitant T2DM published to date showed that SGLT-2i reduces the body weight and WC. The use of SGLT-2i is more effective than other interventions (including other anti-diabetes drugs and placebo) in terms of weight loss ≥ 5%, but not in terms of weight loss ≥ 10%. Other meta-analysis showed that GLP-1 analogues are more effective then SGLT-2i in weight loss in obese patients with T2DM [134].

The dual SGLT-1 and SGLT-2 by licogliflozin in randomized placebo-controlled clinical trials was characterized by mean adjusted percentage changes in body weight after 24 weeks ranging from −0.45% to −3.83%. The rate of adverse reactions with higher licogliflozin dose was clinically significant [140].

Evaluating the results of studies indicated for obesity or resulting in weight reduction and keeping in mind the underlying comorbidities, the selection of pharmacotherapy should be individualized. Such a proposal of therapeutic choice is presented in Figure 1. In addition, taking into account the fact that some of these drugs have a hypoglycemic effect and are used in the treatment of DM, this scheme also includes the recommendations of the American Diabetes Association [141].

## 4. Drugs in Clinical Trials

### 4.1. Cotadutide

Cotadutide is a dual GLP-1/glucagon receptor (GLP-1r/GCGR). In animal models of obesity, the administration of this dual GLP-1R/GCGR agonists resulted in better weight loss, lower glucose levels, and reduced food intake, compared to pure GLP-1r agonists alone [142,143,144]. In humans, on the other hand, a decrease in food intake and a decrease in blood glycemic levels were observed. In Phase 2a clinical trials, however, a significant reduction in the body weight compared to the placebo was observed after just 41 days [145]. In contrast, in Phase 2b, in which overweight or obese patients with concomitant T2DM were studied, a significant decrease in glycated hemoglobin was observed, compared with the control placebo [146]. In overweight individuals without diabetes, dual GLP-1/glucagon infusion increased the energy expenditure to a similar degree as glucagon alone; however, the addition of GLP-1 reduced the hyperglycemic effect of glucagon [147].

### 4.2. Triple GLP-1/Glucagon/GIP Receptor Agonist

The results of studies in animal models of synthetic triple GLP-1/glucagon/GIP receptor agonists are promising. Adding both incretin components to glucagon appears to mitigate the hyperglycemic effects of glucagon better, as compared to the presence of GLP-1 or GIP alone, allowing for a higher dosage of glucagon and, thus, a greater weight loss potential [148]. In the study of SAR441255 by Bossart et al., the weight reduction in mice receiving the triple agonist at 30 μg/kg dose reached approximate 14% after 26 days; in monkeys targeted with a triple agonist at a dose of 10 μg/kg, the weight reduction was approximate 12%. In addition, this compound was more effective than the dual agonist of GLP-1 receptors and glucagon used in the control group [149]. These reports were confirmed with a significant reduction in NASH and NAFLD fatty changes [150]. These reports were confirmed by the research of Knerr et al. [148]. However, there are currently no clinical studies conducted In obese patients [151].

### 4.3. Setmelanotide

Setmelanotide is a synthetic melanocoritin-4 receptor (MCR4) agonist, which is embedded in the leptin-melanocortin pathway. MCR4 is activated by proopiomelanocortin (POMC)-derived neuropeptides, such as α- and β-melanocyte-stimulating hormone (MSH), and plays an important role in hypothalamic body-weight regulation. Its action in the hypothalamus has been found to increase the feeling of satiety, and it can also increase the energy expenditure by enhancing thermogenesis [152]. Current data on the efficacy of setmelanotide originates from a few clinical trials in pediatric patients with rare genetic diseases associated with hyperphagia and obesity. The results of these trials are promising but require further confirmation [153,154]. Phase III studies on this drug to treat rare causes of obesity, such as POMC deficiency, leptin receptor mutation, Prader-Willi syndrome, Bardet-Biedl Syndrome, and Alström syndrome, are currently under way [155]. The MC4R agonist setmelanotide received FDA approval in 2020 for treating subjects with specific genetic defects upstream of MC4R, resulting in obesity. POMC proprotein convertase subtilisin/kexin type 1, or leptin receptor deficiency. Setmelanotide is also being developed in other rare genetic disorders associated with obesity including Bardet-Biedl Syndrome, Alström Syndrome, POMC and other MC4R pathway heterozygous deficiency obesities, and POMC epigenetic disorders [156].

### 4.4. Tesofensine

Tesofensine inhibits the presynaptic transporter of norepinephrine, dopamine, and serotonin. It increases the binding capacity of the dopamine transporter in the dorsal striatum and causes an increase in dopamine concentrations in the nucleus accumbens and in the prefrontal cortex. Its clinical effect is based on the reduction in appetite, as well as the inhibition of the feeling of hunger occurring under the influence of emotions [157,158]. Other studies have also demonstrated an increase in nocturnal metabolism by adipose tissue as a result of the use of tesofensine [159]. In phase II clinical trials, the average weight loss of 2.0% as a result of diet and placebo was observed. Tesofensine at 0.25 mg, 0.5 mg, and 1.0 mg doses combined with diet resulted in an average weight reduction of 4.5%, 9.2%, and 10.6%, respectively. In addition, there was no significant increase in SBP and DBP compared to the placebo, while HR increased by 7.4 beats per minute in the group receiving 0.5 mg of tesofensine [160]. In randomized, placebo-controlled clinical studies, tesofensine was well tolerated, did not affect HR and BP, and resulted in a significant weight loss compared to the placebo in adults with hypothalamic obesity [161].

### 4.5. Methylphenidate

Methylphenidate is an inhibitor of dopamine transport and reuptake. It increases the concentration of dopamine and norepinephrine in the areas of the brain responsible for motivation, reward, attention, and impulsivity. This medicine is currently used in patients with attention deficit hyperactivity disorder (ADHD). Methylphenidate is used for ADHD, during which loss of appetite and weight reduction were observed. In small placebo-controlled studies in children and adolescents conducted to date, methylphenidate has been shown to have a beneficial effect on body weight. The children had greater weight reduction and less weight gain compared with adolescents [162]. Another study demonstrated that methylphenidate significantly increased odor threshold scores compared to the placebo (indicating an improvement in olfactory sensitivity), led to a significantly greater suppression of appetite sensations, and an increased satiety compared to the placebo was observed [163].

### 4.6. Zonisamide with Bupropion

Zonisamide modulates the activity of sodium channels and inhibits carbonate anhydrase, as well as dopaminergic and serotonergic transmission. Its current indication is for epileptic seizures [164]. The initial efficacy of this combination was confirmed by Gadde et al. in a short-term, open-label, preliminary study in which the combination treatment with zonisamide and bupropion resulted in greater weight loss than the treatment with zonisamide alone (an average of 8.5% of baseline weight in the combination group and 3.3% in the zonisamide monotherapy group) [165]. In another study, the authors compared the effectiveness of zonisamide monotherapy at 200 mg and 400 mg doses to the placebo. The change in body weight was −4.0 kg for the placebo, −4.4 kg for 200 mg of zonisamide, and −7.3 kg for 400 mg of zonisamide [166]. A phase II clinical trial has now been completed, but its results have not been published yet [ClinicalTrials.gov Identifier: NCT00339014] [167].

### 4.7. Cetilistat

Cetilistat inhibits pancreatic and gastric lipase; its mechanism of action is the same as that of orlistat [168]. In phase II clinical trials, cetilistat at 80 mg and 120 mg doses three times a day produced a clinical weight reduction effect similar to orlistat at a 120 mg dose three times a day (3.85 kg, 4.32 kg vs. 3.78 kg), while cetilistat at a 40 mg dose had a placebo-like effect (2.94 kg and 2.86 kg, respectively). There was a statistically significant decrease in HbA1c [168,169]. Better tolerance of cetilistat compared to orlistat was also confirmed by Bryson et al. [170].

### 4.8. Type 1 Cannabinoid Receptor Antagonist

The activation of the cannabinoid receptor type 1 (CB1) activates the orexigenic signal (hunger), and its blocking activates the anorexigenic signal (satiety) [171]. The appetite is suppressed by antagonizing the central CB1 receptor. A permeable CB1 receptor antagonist and potentially a reverse agonist rimonabant has been registered for the treatment of obesity. Its effectiveness was confirmed in RCTs, but due to side effects, it has been withdrawn from markets [172,173,174]. Peripheral CB1 Antagonist in obesity research is underway [171]. Studies have demonstrated that peripheral antagonists increase lipolysis and energy expenditure of fat cells, which may initially suggest that in some ways, such a mechanism of action will be effective in obese patients [175]. Another study additionally showed that the peripheral CB1 antagonists inhibit the gut-brain signaling pathway, which may reduce hunger [176]. There are isolated reports of the potential efficacy of peripheral CB1 antagonists in people suffering from Prader-Willi syndrome (with advanced obesity and hyperphagia) [177].

### 4.9. Sildenafil

Sildenafil is a phosphodiesterase type 5 inhibitor. It increases the concentration of cGMP, which leads to an increase in the concentration of nitric oxide, and, consequently, to the relaxation of smooth muscles. It can also lead to increased energy expenditure and increased insulin sensitivity [178,179]. Sildenafil is administered for the treatment of erectile dysfunction and arterial pulmonary hypertension. In animal models, the effectiveness of the chronic use of high doses of sildenafil on exercise and metabolic parameters in obese rats has been demonstrated [180]. One study demonstrated that short-term sildenafil treatment can induce white adipose tissue browning in humans, and this effect can occur through cGMP-dependent protein kinase and mechanistic/mammalian rapamycin (mTOR) signaling pathways [181]. Zemel et al. conducted a controlled, randomized trial of a three-drug therapy with leucine, sildenafil, and metformin. The patients were divided into three groups (full cohorts)—taking a lower dose of sildenafil (1.1 g leucine, 0.5 g metformin, and 0.5 mg sildenafil, respectively) and a higher dose (1.1 g leucine, 0.5 g metformin, 1 mg sildenafil). In addition, two subgroups of patients were distinguished—with AH and hypertriglyceridemia. Dose-dependent weight loss has been demonstrated; the high dose dependent weight loss was observed by 2.4 and 5.0 kg, respectively, in the full cohorts and the high-triglyceride subcohort. The high-dose treatment also lowered SBP by an average of 5.5 mm, with greater effects in people with AH. This combination also reduced the levels of TG and HbA1c significantly [182].

### 4.10. Oxytocin

Oxytocin receptors are located in the pituitary gland, pancreas, adipose tissue and gastrointestinal tract, and hypothalamic neurons that produce oxytocin participate in nerve signaling, primarily in the reward system and the hypothalamic centers of satiety and hunger [183]. In studies on men, oxytocin has been shown to reduce calorie intake with a preferential effect on fat intake. It also increases the level of the anorexigenic hormone cholecystokinin, without affecting the appetite or other hormones that regulate appetite. It results in a shift from the utilization of carbohydrates to the metabolism of fats and improves insulin sensitivity [184]. Other studies have shown that the use of oxytocin affects the morning feeling of hunger in obese men, showing a difference in the obtained anorexigenic effect between obese men and those with a BMI within the normal limits [185]. One functional magnetic resonance imaging study demonstrated that, after the intranasal administration of oxytocin, compared with the placebo, participants proved significantly impaired functional connectivity between the ventral tegmental area (VTA) and the insula, oral sensory-somatic cortex, amygdala, hippocampus, tegmentum, and medial occipitotemporal gyrus in response to looking at high-calorie foods. There was no difference in the functional connectivity between the VTA and these brain areas when comparing oxytocin and the placebo for low-calorie food images and non-food images. This may in some way explain the anorexigenic effect of oxytocin—by reducing the drive towards high-calorie foods [186]. Some studies also suggest the effectiveness of oxytocin in the suppression of behavioral impulses [187]. Espinoza et al. demonstrated that, in patients with sarcopenic obesity, oxytocin led to a significant increase (by 2.25 kg) in lean body mass compared to the placebo, with a simultaneous tendency to reduce the fat mass and significantly reduce the plasma LDL-c cholesterol levels (by 19.3 mg/dl) compared to the placebo. There were no significant changes in BMI, appetite assessment, glycemic levels, plasma HDL-C concentrations, TG, or the presence of depressive symptoms [188]. However, studies with oxytocin are usually conducted in small male populations, as no differences compared to placebo were observed in women without eating disorders. In the pubMed database, there are few clinical studies evaluating the effect of intranasally administered oxytocin on food consumption in women. One of them showed a reduction in the intake of meals under stress in women [189]. In studies on patients with Prader-Willi syndrome, intranasal oxytocin has been shown to have a smaller effect than the placebo on the number of hyperphagy incidents [55]. In contrast, another study found that the type of genetic disorder, gender, and age may be relevant in the occurrence of positive effects in the context of hyperphagia attacks and social behavior, compared to the placebo [190]. Undoubtedly, the group of patients with Prader-Willi syndrome require further research and analysis, due to the ambiguity of the results obtained so far [191,192,193].

### 4.11. Velneperit

Velneperit is a type 5 neuropeptide Y receptor antagonist. The binding of neuropeptide Y to the receptor leads to a decrease in the feeling of hunger [194]. In phase II clinical trials, patients receiving velneperit 800 mg had a reduced body weight by an average of 3.8 kg, compared to 0.8 kg with the placebo. In the 800 mg group, 35% of the patients reduced their body weight by more than 5%, while in the placebo group, only 12% showed a >5% reduction. At a dose of 1600 mg of velneperit, a higher weight reduction (7.1 kg) was demonstrated, compared to the placebo (4.3 kg) [195,196]. A phase II study to evaluate the efficacy and safety of the combination of velneperit and orlistat is currently under way [ClinicalTrials.gov Identifier: NCT01126970].

### 4.12. Amylin Analogues

Amylin is a peptide secreted with insulin by the pancreatic β cells. It has been proven to be responsible for reducing postprandial glucose, slowing down gastric emptying, and increasing satiety sensation [197]. Amylin is also secreted in the stomach, sympathetic ganglia, and centrally in the hypothalamus, stimulating the feeling of satiety, and acting in the ventral tegmental area and nucleus accumbens, it affects the food intake associated with the reward aspect [197,198]. In addition, amylin has a synergistic effect with leptin in reducing food intake. On the other hand, amylin can aggregate and form amyloid deposits in the β cells of the pancreatic islets. This process, which probably stimulates hyperglycemia, leads to the progressive degradation of the pancreatic islets and, consequently, the development of T2DM [197,199]. Studies in animal models have confirmed the positive effect of amylin analogues (some of the substances used are double agonists of calcitonin and amylin receptors) on weight reduction (up to 10%) and the improvement of glucose metabolism [200,201,202]. Currently, pramlintide is the only amylin analogue registered (for the treatment of patients with DM) [203,204]. In a double-blind, placebo-controlled, 4 month study by Smith et al., a weight reduction of approximate 3% was obtained [205]. In their previous randomized trial, a six-week pramlintide therapy led to approximately 2% weight loss [206]. Research by Aronee et al. confirmed these data. In a 16-week randomized trial, the weight loss was approximately 3.5% [207]. Although these results are insignificant, it should be noted that in studies on combination therapies, such as pramlintide with sibutramine, or pramlintide with phentermine, the effectiveness of weight reduction increases to approximately 11% [208]. Another amylin agonist is cagrilintide, administered subcutaneously once a week. In the phase two study, the weight reduction was up to about 10%; it was also shown to be slightly more effective compared to liraglutide (4.5 mg of cagrilintide—10.8% vs. 3.0 mg of liraglutide—9%) [209]. In the combination therapy of cagrilintide with semaglutide (4.5 mg + 2.4 mg), the weight reduction was approximately 17% [210].

### 4.13. Peptide Tyrosine-Tyrosine

Peptide tyrosine-tyrosine is secreted by the L cells of the ileum and colon. In response to the food intake, it stimulates the feeling of satiety and inhibits hunger in the hypothalamic centers by activating the NPY2R receptor (neuropeptide Y receptor type 2) [211,212]. It was also found to slow down gastric emptying, inhibit the secretion of stomach acid, exocrine pancreatic function, and insulin [213]. Studies in animal models confirm the effectiveness of PYY analogues, both in weight and food intake reduction [214,215]. Studies on humans conducted to date do not allow for a clear assessment of the effectiveness of PYY in weight reduction yet. In a 12-week, randomized study, intranasal administration of 200 or 600 μg of PYY three times a day before each meal in obese patients (BMI 30–43 kg/m^2^), no significant weight loss demonstrated [216].

### 4.14. Fibroblast Growth Factor Analogue 21

Fibroblast growth factor 21 (FGF-21) is produced in the liver, adipose tissue, skeletal muscle, and pancreas [217,218]. In the white adipose tissue, it stimulates glucose uptake and secretion of adiponectin, and in the brown adipose tissue, it results in glucose uptake and thermogenesis, and peripherally it increases insulin sensitivity of the tissues. This is due to an increase in adiponectin secretion [217,219]. In the liver, it inhibits the action of the growth hormone, regulates the oxidation of fatty acids on an empty stomach and after a meal, and, thus, has a positive effect on the lipid profile [220]. In people with NAFLD, obesity, and T2DM, FGF-21 concentrations have been found to be elevated, suggesting that there may be resistance to its effects [221]. In animal models, the effectiveness of the FGF-21 analogue in reducing body weight was 3–15%, depending on the dose [222]. In the study by Kaufman et al., the administration of the long-acting fusion protein immunoglobulin 1 and FGF-21 (Ig1) Fc-FGF-21 (AKR-001) to T2DM patients with a BMI of 25–40 kg/m^2^ and An HbA1c of 6.5–10% or lower has not been shown to have a significant effect on weight reduction, as well as on FPG, FINS, fasting plasma glucagon, C-peptide, HOMA-IR, HDL-C, non-HDL-C, and TG [223]. In turn, in the study of Baruch et al. over the double antibody of the FGF-21 receptor and the klotho protein β (BFKB8488A), the weight reduction on the eighth day of the drug was approximately 2%, depending on the dose. In addition, a single administration of this drug was demonstrated to result in dose-dependent and metabolically beneficial effects, as evidenced by an increase in serum adiponectin concentrations similar to HDL-C and a decrease in TG, LDL-C, and FINS concentrations. Notably, these effects were prolonged, reaching up to 60 days for several cohorts receiving high doses [222]. Similar conclusions were drawn in the study by Sanyal et al. of pegbelfermin, which is a pegylated human FGF-21 (BMS-986036). In 16-week therapy with this drug in patients with T2DM, with a BMI at least 25 kg/m^2^ and biopsy-confirmed NASH, despite a significant decrease in the absolute hepatic fat fraction, no significant weight reduction was observed with both 10 and 20 mg daily doses [224].

### 4.15. Mirabegron

Mirabegron is a selective β3 adrenergic receptor agonist registered for the treatment of overactive bladder [225,226]. The β3 adrenergic receptors are found mainly in the myocardium, smooth muscle, and adipose tissue, and their activation increases lipolysis in white adipose tissue and thermogenesis in brown adipose tissue [227,228]. Lipids are stored and released in the white adipose tissue, and, additionally, perform the function of an endocrine gland, secreting adipokines, such as adiponectin and leptin. In obesity, there is an overgrowth of white adipocytes, followed by fibrosis, necrosis, and infiltration of the immune cells, which leads to local and systemic inflammatory conditions, IR, and metabolic dysfunction. In turn, brown adipose tissue uses glucose and lipids to produce heat mediated by thermogenin (UCP1). Brown adipose tissue is present in humans, but with age and weight gain, its production decreases. Mirabegron, by UCP1 induction, stimulates the so-called browning of the white adipose tissue, as a result of which it becomes capable of thermogenesis [229,230]. During mirabegron treatment, the activation of the brown adipose tissue and energy expenditure has been observed to increase, and the carbohydrate and lipid metabolism to improve. In addition, there is a reduction in the content of adipose tissue and the concentration of pro-inflammatory cytokines in the body [227,229]. Despite the suggested mechanisms of action of this drug, it has not proven to reduce weight significantly. Although in an animal model the use of 12-week therapy with this drug was associated with a reduction of 34% and 54% and an increase of 35% in the epididymal fat, the LDL-C and HDL-C levels did not affect the body weight and FPG levels [231]. In another study of a small group of young females with an average BMI of about 25 kg/m^2^, 4 weeks of mirabegron 100 mg per day, despite an increase in the BAT metabolic activity and a beneficial effect on the level of HDL-C, ApoA1, adiponectin, insulin sensitivity, glucose effectiveness, and insulin secretion, this did not cause a significant body weight loss [232].

## 5. Conclusions

Obesity is a civilization disease associated with a number of life-threatening consequences. Therefore, new therapeutic solutions, involving not only better effectiveness but also additional pleiotropic activities, are continually being sought. Due to the growing number of drugs that reduce body weight, the therapeutic choice is becoming more and more difficult. Therefore, when deciding on a specific medication, we must take into account not only its effectiveness, but also the patient’s preferences and concomitant diseases. The main side effects of these drugs, their interactions, and contraindications are presented in Table 6 (based on clinical trial data and Summary of Product Characteristics). Taking into account the increased risk of selected adverse effects in diabetic patients, it seems reasonable to perform diagnostic tests for disorders of carbohydrate metabolism before applying treatment for weight reduction. In addition, it is worth carefully analyzing the needs of patients, and the severity of side effects, in order to obtain the best therapy compliance.

## Figures and Tables

**Figure 1 healthcare-11-00433-f001:**
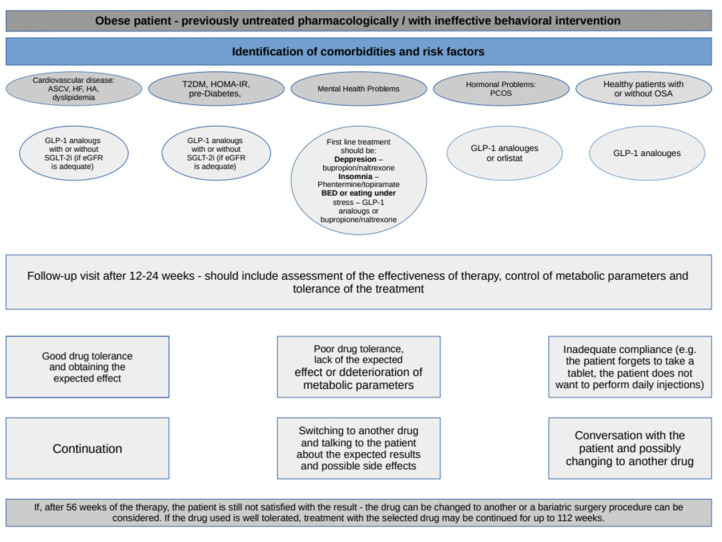
Selection of a specific pharmacological agent for a particular patient [26,27,52,53,54,55,62,63,64,73,74,75,76,89,90,91,101,102,103,104,105,136,138,141].

**Table 1 healthcare-11-00433-t001:** Clinical studies assessing the efficacy bupropion and naltrexone combination.

Study	Study Design	Baseline Sample	Duration	Dose of Bupropion/Naltrexon	Body Mass Reduction (%)	Percent of Patients with 5/10/15% Weight Reduction	Additional Benefits
COR I [52]	RCT	1742 patients ages 18–65 years with a BMI of 30–45 kg/m^2^ and uncomplicated obesity or BMI 27–45 kg/m^2^ with dyslipidaemia or AH, or both	56 weeks	360 mg/32 mg 360 mg/16 mg	6.1 5.0	48/25/12 39/20/9	SSR in WC, TG, hs-CRP, FINS, FPG, HOMA-IR, SBP, DBP. SSI in HDL-C. SSR in WC, TG, hsCRP, HOMA-IR, SBP, DBP. SSI in HDL-C.
COR II [53]	RCT	1496 patients ages 18–65 years with a BMI of 30–45 kg/m^2^, or a BMI of 27–45 kg/m^2^ and controlled AH and/or dyslipidemia.	56 weeks	360 mg/32 mg	6.5	50.5/28.3/13.5	SSR in WC, TG, LDL-C, hsCRP, FINS, HOMA-IR, SBP. SSI in HDL-C.
COR-BMOD [54]	RCT	793 patients ages 18–65 years with a BMI of 30–45 kg/m^2^, or a BMI of 27–45 kg/m^2^ in the presence of controlled AH and/or dyslipidemia	56 weeks	360 mg/32 mg	9.3	66.4/41.5/29.1	SSR in WC, TG, FINS, HOMA-IR. SSI in HDL-C.
COR-DM [55]	RCT	505 patients with T2DM and ages 18–70 years, with a BMI of ≥27 and ≤45 kg/m^2^, HbA1c between 7% and 10%, and FPG <270 mg/dL	56 weeks	360 mg/32 mg	5.0	≥5% weight loss in 44.5% patients	SSR in WC, HbA1c, TG. SSI in HDL-C.

BMI—body mass index, COR I- Contrave Obesity Research I; COR II—Contrave Obesity Research II, COR—BMOD—Contrave Obesity Research in adjunct to intensive behavioral modification, COR-DM—Contrave Obesity Research Diabetes Study Group, DBP—diastolic blood pressure, FINS —fasting serum insulin, FPG—fasting plasma glucose, AH—arterial hypertension, HDL-C—high-density lipoprotein cholesterol, HOMA-IR—homeostatic model assessment for insulin resistance, hsCRP—high sensitivity C-reactive protein, LDL-C—low-density-lipoprotein cholesterol, RCT—randomized controlled trials, SBP—systolic blood pressure, SSI—statistical significant increase, SSR—statistical significant reduction, T2DM—Type 2 Diabetes Mellitus, TG—triglycerides, WC—waist circumference.

**Table 2 healthcare-11-00433-t002:** Clinical trials evaluating the efficacy of the combination of phentermine and topiramate.

Study	Study Design	Baseline Sample	Duration	Dose of Phentermine/Topiramat	Body Mass Reduction (%)	Percent of Patients with 5/10/15% Weight Reduction	Additional Benefits
EQUIP [62]	RCT	1267 patients with a BMI of ≥35 kg/m^2^, TG ≤ 200 mg/dL with treatment of 0–1 lipid-lowering medication, BP ≤ 140/90 mmHg with treatment of 0–2 antihypertensive medications, FPG ≤ 110 mg/dL	56 weeks	15 mg/92 mg 3.73 mg/23 mg	10.9 5.1	66.7/47.2/32.3 44.9 /18.8/7.3	SSR in WC, TG, FPG, TCH, LDL-C, SBP, DBP, TCH/HDL-C ratio. SSI in HDL-C. SSR in WC, SBP. SSI in HDL-C.
CONQUER [63]	RCT	2487 patients aged 18–70 years, with a BMI of 27–45 kg/m^2^, ≥2 comorbidities at baseline (SBP 140–160 mmHg or 130–160 mmHg in T2DM patients, DBP 90–100 mmHg or 85–100 mmHg in T2DM patients or taking ≥2 antihypertensive drugs; TG 2.26–4.52 mmol/L or using ≥2 lipid-lowering drugs; FBG > 5.5 mmol/L or 2 h after OGTT > 7.77 mmol/L or diagnosed T2DM managed with lifestyle changes or metformin monotherapy; WC ≥ 102 cm for men or 88 cm for women.	56-weeks	15 mg /92 mg 7.5 mg/46 mg	9.8 7.8	70/48/- 62/37/-	SSR in WC, SBP, DBO, TCH, LDL-C, TG, FPG, HbA1c, FINS, Homa-IR, hsCRP. SSI in HDL-C, adiponectin. SSR in WC, SBP, TCH, TG, FPG, HbA1c, FINS, HOMA-IR, hsCRP. SSI in HDL-C, adiponectin.
SEQUEL [64]	RCT	676 patients with a BMI of 27–45 kg/m^2^ and two or more comorbidities (AH, dyslipidaemia, diabetes or prediabetes, or abdominal obesity),	56 weeks	15.0 mg/92 mg 7.5/46 mg	10.5 9.3	79.3/53.9/31.9 75.2/50.3/24.2	SSR in TG, LDL-C, FPG, FINS, HbA1c, SBP, DBP. SSI in HDL-C. SSR in TG, LDL-C, FINS, HbA1c, SBP, DBP. SSI in HDL-C.
EQUATE Clinicaltrials.gov: NCT00563368	RCT	756 patients ages 18–70 years of age who had a BMI of ≥30 and ≤45 kg/m^2^	28 weeks	15 mg/92 mg 7.5 mg/46 mg	9.2 8.5	68/42/- 64/40/-	SSR in WC, SBP, HBA1c. SSR in WC, HbA1c, adiponectin.

BMI—body mass index, DBP—diastolic blood pressure, FINS—fasting serum insulin, FPG—fasting plasma glucose, AH—arterial hypertension, HDL-C—high-density lipoprotein cholesterol, HOMA-IR—homeostatic model assessment for insulin resistance, hsCRP—high sensitivity C-reactive protein, LDL-C—low-density-lipoprotein cholesterol, RCT—randomized controlled trials, SBP—systolic blood pressure, SSI—statistical significant increase, SSR—statistical significant reduction, T2DM—Type 2 Diabetes Mellitus, TG—triglycerides, WC—waist circumference.

**Table 3 healthcare-11-00433-t003:** Clinical studies evaluating the effectiveness of the combination of lorcaserin.

Study	Study Design	Baseline Sample	Duration	Dose of Lorcaserine	Body Mass Reduction (%)	Percent of Patients with 5/10/15% Weight Reduction	Additional Benefits
BLOM [76]	RCT	3182 patients ages 18–65 years and a BMI of 30–45 kg/m^2^ or 27–45 kg/m^2^ with at least 1 coexisting condition (AH, dyslipidemia, cardiovascular disease, impaired glucose tolerance, or sleep apnea).	52 weeks	10 mg BID	5.81	47.5/22.6	SSR in SBP, DBP, TCH, LFL-C, TG, FPG, FINS, HOMA-IR, hsCRP, fibrinogen.
BLOOSOM [73]	RCT	4008 patients ages 16–65 years, with a BMI of 30–45 kg/m^2^ or 27–29.9 kg/m^2^ with obesity-related comorbid condition	52 weeks	10 mg BID 10 mg QD	5.8 4.7	47.2/22.6/- 40.2/17.4/-	SSR in WC, TG, ApoB, total body fat, lean body mass. SSI in HDL-C. SSR in WC, TG/ SSI in HDL-C.
BLOOM-DM [74]	RCT	604 patients ages 18–65 years with T2DM treated with metformin, a SFU, or both; with HbA1c 7–10%; with a BMI of 27–45 kg/m^2^	52 weeks	10 mg BID 10 mg QD	4.7 5.0	37.5/16.3/- 44.7/18.1/-	SSR in WC, HC, HR, FPG, HbA1c, HOMA-IR. SSI in HDL-C. SSR in HC, HR, apoA1, FPG, FINS.
CAMELLIA -TIMI 61 [75]	RCT	12,000 patients ages at least 40 years old with a BMI of ≥27 kg/m^2^ with either established atherosclerotic cardiovascular disease or multiple cardiovascular risk factors	3.3 years	10 mg BID	4.2	38.7/14.6	SSR in HbA1c, FPG.

BID—twice daily, BMI—body mass index, DBP—diastolic blood pressure, FINS—fasting plasma insulin, FPG—fasting plasma glucose, AH—arterial hypertension, HDL-C—high-density lipoprotein cholesterol, HOMA-IR—homeostatic model assessment for insulin resistance, hsCRP—high sensitivity C-reactive protein, LDL-C—low-density-lipoprotein cholesterol, QD—once daily, RCT—randomized controlled trials, SBP—systolic blood pressure, SSI—statistical significant increase, SSR—statistical significant reduction, SFU—sulfonylurea, T2DM—Type 2 Diabetes Mellitus, TG—triglycerides, WC—waist circumference, Incretinomimetics—GLP-1 analogues.

**Table 4 healthcare-11-00433-t004:** Clinical studies assessing the efficacy of liraglutide therapy in obese or overweight patients.

Study	Study Design	Baseline Sample	Duration	Dose of Liraglutide (9 mg)	Body Mass Reduction (%)	Percent of Patients with 5/10/15% Weight Reduction	Additional Benefits
SCALE Obesity and Prediabetes [89]	RCT	3731 patients ages ≥ 18 years with a BMI of ≥30 kg/m^2^ or ≥27 kg/m^2^ with treated or untreated dyslipidemia or AH, without DM	56 weeks	3.0	8.0	63.2/33.1/14.4	SSR in SBP, DBP, HbA1c, FPG, IR, and beta-cell function. SSI in FINS, C-peptide, HDL-C, PAI-1, adiponectin.
SCALE Diabetes [90]	RCT	846 patients ages ≥18 years with a BMI of ≥27 kg/m^2^ taking 0–3 oral hypoglycemic agents (metformin, thiazolidinedione, sulfonylurea) with stable body weight, and HbA1c 7–10%.	56 weeks	1.8 3.0	4,7 6,0	40.4/15.9/- 54.3/25.2/-	SSR in WC, HbA1c, FPG, PPG, HOMA-B. SSI in glucagon, proinsulin, proinsulin to insulin ratio, hsCRP. SSR in WC, HbA1c, FPG, PPG, HOMA-B, HOMA-IR, SBP, TCH, VLDL, TG. SSI in HDL-C, glucagon, proinsulin, proinsulin to insulin ratio, fibrinogen, PAI-1, hsCRP.
SCALE Maintenance [92]	RCT	422 patients ages ≥ 18 years with a stable BMI of ≥30 kg/m^2^ or ≥27 kg/m^2^ with comorbidities of treated or untreated dyslipidemia and/or treated or untreated AH who lost ≥5% of initial weight during a low-calorie diet run-in, without DM	56 weeks	3.0	6.2	50.5/26.1/-	SSR in WC, HbA1c, FPG, FINS, TG, hsCRP, SBP.
SCALE Sleep Apnea [91]	RCT	359 patients ages 18–64 years with a stable BMI (<5% change during the previous 3 months) and a BMI of ≥30 kg/m^2^ with moderate or severe OSA	32 weeks	3.0	5.7%	46.3/23.4/-	SSR in HbA1c, FPG, SBP.

BMI—body mass index, DBP—diastolic blood pressure, DM—diabetes mellitus, FINS—fasting serum insulin, FPG—fasting plasma glucose, AH—arterial hypertension, HDL-C—high-density lipoprotein cholesterol, HOMA-IR—homeostatic model assessment for insulin resistance, HOMA-B—homeostatic model assessment—beta, hsCRP—high sensitivity C-reactive protein, LDL-C—low-density-lipoprotein cholesterol, RCT—randomized controlled trials, OSAS—obstructive sleep apnea, SBP—systolic blood pressure, SSI—statistical significant increase, SCALE—Safety and Clinical Adiposity Liraglutide Evidence, SSR—statistical significant reduction, T2DM—Type 2 Diabetes Mellitus, TG—triglycerides, WC—waist circumference.

**Table 5 healthcare-11-00433-t005:** Clinical studies assessing the efficacy of semaglutide therapy in obese or overweight patients.

Study	Study Design	Baseline Sample	Duration	Dose of Semaglutide (mg)	Body Mass Reduction (%)	Percent of Patients with 5/10/15% Weight Reduction	Additional Benefits
STEP 1 [101]	RCT	1961 patients ages ≥ 18 years old with one or more self-reported unsuccessful dietary efforts to lose weight and either a BMI of ≥30 kg/m^2^ or ≥27 kg/m^2^ with ≥1 treated or untreated weight-related coexisting conditions (i.e., AH, dyslipidemia, OSAS, CVD)	68 weeks	2.4	14.9	86.4/69.1/50.5	SSR in WC, SBP.
STEP 2 [102]	RCT	1210 patients ages ≥ 18 years old, with at least one reported unsuccessful dietary effort to lose weight, a BMI of ≥27 kg/m^2^, HbA1c 7–10%, T2DM at least 180 days before screening	68 weeks	2.4 1.0	9.6 6.99	68.8/45.6/25.8 57.1/28.7/13.7	SSR in WC, HbA1c, SBP, FSIN, DBP, TCH, VLDL, free fatty acids, TG, CRP. SSI in HDL-C. SSR in HbA1c.
STEP 3 [103]	RCT	611 patients with ages ≥ 18 years old, reported ≥1 unsuccessful dietary efforts to lose weight and BMI ≥30 kg/m^2^ or ≥27 with ≥1 weight-related comorbidity (CVD, dyslipidemia, AH, OSAS	68 weeks	2.4	16	86.6/75.3/55.8	SSR in WC, SBP, DBP, FPG, HbA1c, FSIN, TCH, LDL-CVLDL, free fatty acids, TG, CRP.
STEP 4 [104]	RCT	902 patients with ages ≥ 18 years old, with ≥1 reported unsuccessful dietary efforts to lose weight and a BMI of ≥30 kg/m^2^ or ≥27 with ≥1 weight-related comorbidity (CVD, dyslipidemia, AH, OSAS) without T2DM.	68 weeks	2.4	7.9	88.7/79/63.4	SSR in WC, SBP, HfA1c, FSIN, FPG, TCH, LDL-C, VLDL-C, TG. SSI in
STEP 6 [105]	RCT	401 patients with ages ≥ 18 years old, with ≥1 reported unsuccessful dietary efforts to lose weight and a BMI of ≥27 kg/m^2^ ported ≥2 treated or untreated weight-related comorbidity or ≥35 with ≥1 weight-related comorbidity (dyslipidemia, AH, or, in Japan only, T2DM diagnosis 80 days before screening and HbA1c 7–10%).	68	2.4 1.7	13.2 9.6	83/61/41 72/42/24	SSR in WC, HbA1c, FSIN, SBP, DBP, PAI-1, CRP, TCH, LDL-C, TG FFA. SSRI in WC, HbA1c, FSIN, SBP, DBP, PAI-1, CRP, FFA, TG, TCH, LDL-C.

BMI—body mass index, DBP—diastolic blood pressure, DM—diabetes mellitus, FINS—fasting serum insulin, FPG—fasting plasma glucose, AH—arterial hypertension, HDL-C—high-density lipoprotein cholesterol, HOMA-IR—homeostatic model assessment for insulin resistance, HOMA-B—homeostatic model assessment-beta, hsCRP—high sensitivity C-reactive protein, LDL-C—low-density-lipoprotein cholesterol, RCT—randomized controlled trials, OSAS—obstructive sleep apnea, SBP—systolic blood pressure, SSI—statistical significant increase, SCALE—Safety and Clinical Adiposity Liraglutide Evidence, SSR—statistical significant reduction, T2DM—Type 2 Diabetes Mellitus, TG—triglycerides, WC—waist circumference.

**Table 6 healthcare-11-00433-t006:** Summary of the most common and serious side effects, interactions, contraindications, and dosage of preparations used in the treatment of obesity [26,27,52,53,54,55,62,63,64,73,74,75,76,89,90,91,101,102,103,104,105,136,138].

Drug	Most Common Side Effects	Serious Side Effects	Interaction	Contraindications	Dosage and Route of Administration
Orlistat	Abdominal pain and discomfort, loose stools, liquid stools, push to the stools, rectal pain and discomfort, headache, fatigue, anxiety	Hypoglycemia infections	Cyclosporine, Acarbose, Oral antidiabetic drugs, Fat-soluble vitamins, Amiodarone	Hypersensitivity to orlistat or to any of the excipients, chronic malabsorption syndrome, cholestasis, breast-feeding	3x/day, p.o.
Bupropion /naltrexone	Nausea, constipation, vomiting, dizziness, dry mouth, headache	Affective disorder, suicidal ideation, seizure, cholecystitis, hepatitis, erythema multiforme, Stevens-Johnson syndrome	Active substances metabolized by cytochrome P450 isoenzymes; CYP2D6 isoenzyme substrates; inducers, inhibitors and substrates of the CYP2B6 isoenzyme; organic cation transporter 2 inhibitors	Hypersensitivity to bupropion, naltrexone or to any of the excipients; uncontrolled hypertension; epilepsy or history of seizures; cancer tumor in the central nervous system; the period immediately after abrupt withdrawal from alcohol or benzodiazepines in an addicted person; history of bipolar affective disorder; psychic bulimia or anorexia nervosa; dependence on long-term use of opioids or opiate agonists (e.g., methadone) and the period immediately after abrupt opiate withdrawal in an addicted person; taking monoamine oxidase inhibitors; severe hepatic impairment; end-stage renal failure or severe renal impairment	2x/day, p.o
Phentermine/topiramat	Dry mouth, paresthesia, constipation, memory impairment insomnia, depression, anxiety	Metabolic acidosis, hypokalemia, angle-closure glaucoma, transient blindness, deafness, atrial fibrillation, arrhythmias, deep vein thrombosis, falls	Antiepileptic drugs, Hydrochlorothiazide, Hypericum perforatum, oral contraceptives, metformin, pioglitazone, glibenclamide, digoxin, lithium, risperidone, central nervous system depressants, carbonic anhydrase inhibitors, potassium-sparing diuretics, valproic acid	Hypersensitivity to phentermine, topiramate, or to any of the excipients; pregnancy; breast feeding; treatment with monoamine oxidase inhibitors	2x/day, p.o
Liraglutide	Gastrointestinal symptoms (for example: nausea, vomiting, diarrhea, constipation, dry mouth, dyspepsia), insomnia, headache, hives	Hypoglycemia, inflammation of the pancreas, cholecystitis, cholelithiasis, acute renal failure	Paracetamol, atorvastatin, griseofulvin, digoxin, lisinopril, oral contraceptives	Hypersensitivity to liraglutide or to any of the excipients, medullary thyroid cancer, pregnancy, breast feeding	1x/day, s.c.
Semaglutide	Gastrointestinal symptoms (for example: vomiting, diarrhea, constipation, nausea, abdominal pain, dyspepsia, bouncing), headache, dizziness, tiredness	Cholelithiasis, hypoglycemia	Paracetamol, oral contraceptives	Hypersensitivity to semaglutide or to any of the excipients, medullary thyroid cancer, pregnancy, breast feeding	1x/week, s.c.
Lorcanserin	Nausea, vomiting, constipation, diarrhea, fatigue, upper respiratory tract infection, urinary tract infection, back pain, headache, dizziness, rash, attention and memory deficit, priapism, hyperprolactinemia	Infections, serotonin syndrome, hypoglycemia	Drugs that interfere with serotonin neurotransmission, drugs metabolized by cytochrome P450 2D6	Hypersensitivity to lorcaserin or to any of the excipients, pregnancy, breast feeding	2x/day, p.o.

p.o.—orally, s.c.—subcutaneously.

## Data Availability

No new data were created or analyzed in this study.

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
