# Peer review of "Pharmacological Support for the Treatment of Obesity—Present and Future"

_healthcare, 2023, doi:10.3390/healthcare11030433_

Round 1
Reviewer 1 Report
The paper summarizes the effectiveness of available drugs in the treatment of obesity. Each molecule is described in terms of its mechanism of action, indications for its use, and contains tables summarizing the results of studies using it in the treatment of obesity. Only the description of dulaglutide requires a more extensive description of the studies. The value of the work is also the description of new drugs, not yet used in everyday clinical practice.
Author Response
Dear reviwer,
Thank you very much for all your valuable comments and suggestions.
I took into account your suggestion - I developed the part about dolaglutide. I also searched databases of medical articles, but despite this, I did not find any new publications on the use of dulaglutide in the treatment of obesity.
Best regrates
Kacper Deska
Reviewer 2 Report
The paper entitled “Pharmacological support for the treatment of obesity – present and future”, includes potentially relevant data for the clinical aspects of pharmacology directed to endocrinology, diabetology and potentially for the development and application of personalized therapeutic algorithm focused on the management of obesity. The Authors of this publication tend to characterize chosen facts describing the: ”….weight-reducing drugs…” as well as effectiveness and individualization of use mentioned substances.
Remarks:
1. The Authors need to prepare a table – in which they must consider the most common effects, interactions and contraindications – in relation to the currently used active substances described in the publication.
2. The Authors must prepare a figure graphically depicting the mechanisms and gripping points of the active substances described in the publication.
3. The Authors in the last sentence of Introduction section (“…Therefore, the purpose of the following review is to summarize the latest knowledge about weight-reducing drugs, taking into account their effectiveness and individualization of use…”; lines 78-80) mentioned about individualization of use – therefore, the Authors need to make an attempt – perhaps also in tabular form – to determine – at least for active substances currently used – in which age group / age groups the substances (considering their mechanism of action) can be applied.
4. The Conclusion section does not contain practical guidelines that an endocrinologist, diabetologist or family doctor could follow in choosing the active substances described by the authors – Conclusion section need to be modified.
5. The Authors of the Publication use Polish words/Polish language, e.g. Table no. 1 ”...Dawka...”, “...Dodatkowe istotne korzyÅ›ci...”; Table no. 2 “...Dawka phenformina...”, “...Dodatkowe istotne korzyÅ›ci...”; Table no.3 “...Dawka lorcaseryny...”, “... Dodatkowe istotne korzyÅ›ci... “
6. The authors of the paper in Table no. 2 refer to the active substance - phenformin. It should also be added that: “...The use of phenformin was discontinued in the United States in 1976 because of probable association with lactic acidosis…” (Gan SC, Barr J, Arieff AI, Pearl RG. Biguanide-associated lactic acidosis Case report and review of the literature. Arch Intern Med. 1992 Nov;152(11):2333-6. doi: 10.1001/archinte.152.11.2333. PMID: 1444694.).
7. Figure no.1 (Selection of a specific pharmacological agent for a particular patient.) may raise doubts. A question should be asked whether the Figure no. 1 was independently developed and prepared by the Authors (the middle part of the Figure no. 1 – a white magnifying glass on a red background).
If the Figure (no.1) was prepared by the Authors themselves – then the sources (scientific publications) on the basis of which the Authors developed the Figure (no.1) in question should be indicated.
Author Response
Dear Reviewer,
Thank you very much for all your valuable comments and suggestions. Below I have described the corrections made in the text.
2# The mechanisms of action of all substances are explained in the main text of the article. In our opinion, the inclusion of additional diagrams repeating the mechanisms of action of the substance goes beyond the scope of the article and would only constitute a significant extension of the volume of the manuscript without substantive value.
1# + 3# + 4# The conclusions section has been supplemented. We have also added tables with the most common side effects, serious side effects, interaction, contraindications and dosage of drugs registered in the treatment of obesity. Thanks to this, it is a continuation of the conclusions section. In our opinion, the whole article contains a lot of valuable practical tips.
5# Corrections have been made.
6# The entire chapter as well as the table to it concerned the combination of phentermine with topiramate. There was a translation error in one place in the table, which has been corrected.
7# The figure is ours, The watermark that was visible in the previous version of the manuscript was only an editorial error - we sincerely apologize for it. This has been corrected.
Best regrates
Kacper Deska
Reviewer 3 Report
Thank you for your hard work on this valuable topic. I have reviewed the manuscript and have many suggestions for you to consider. I hope I am clear with my comments and wish you the best. Happy Holidays.

Author Response
Dear Reviewer,
Thank you for all the suggestions and I sincerely apologize for some editorial and translation oversights. I took into account all the suggested corrections in the work.
I have corrected the language, removed redundant words, developed abbreviations and corrected the ones used (following your suggestions). For the sake of transparency of the corrections, I attach a version in which I have highlighted the corrections in yellow. Plus some of my comments
line 166 - throughout the test we changed "HA - hypertonia arterialis" to "AH- arterial hypertension" (HA is speld out in the line #44 now)
line 239 - "BED" is sped out in the line #226
lines 234 to 240 - I divided the paragraph into shorter sentences and eliminated redunded words
lines 448-450 - I changed this part to make it easier to read (now lines 433-435)
line 561 - I change "flozins" to SGLT2i in whole text
line 569 - I delated this part about GLP-1 agonist to be more to be more understandable
line 697 - I change the sentence to "Peripheral CB1 antagonist in obesity research in underway" - now line 678/679
line 711 - I add indications for the use of sildenafil
Best regrates,
Kacper Deska

Round 2
Reviewer 2 Report
The authors of the publication modified the text of the article submitted to the editorial office.
In its present form, the article is ready for publication.
Author Response
Dear reviewer
Thank you very much for your positive review.
Best wishes
Kacper Deska
Reviewer 3 Report
Thank you for your work on the reviewers comments. I have made few more minor comments for clarification.
Good luck.

Author Response
Dear Reviewer,
thank you very much for all your valuable comments and suggestions. I added all the language suggestions in the text (the numbering of the paragraphs changed after the previous correction). In the attachment I add the revised version of the manuscript - in green I underlined the suggestions from the second answer. I have no comments to them and I have adapted to all of them.
Best wishes
Kacper Deska
